# Neural Networks with Adaptive Activation Functions and their Application to the Solution of PDEs

## Abstract

We study fully connected neural networks for function approximation, focusing on the role of adaptive activation functions. Recent work has shown that introducing trainable parameters into activation functions, particularly rational functions, can substantially improve network expressiveness. We extend this idea to sampling-based neural networks, which replace backpropagation with forward sampling to achieve faster training. Existing sampling-based methods, however, only employ fixed activation functions, limiting their performance. To address this gap, we develop a computational framework that integrates adaptive activation functions into sampling-based training, enabling direct learning of activation parameters through sampling without gradient-based optimization. Experiments demonstrate that our approach preserves the efficiency of sampling-based methods while significantly improving approximation accuracy, highlighting the benefits of parameterized activation functions in non-gradient training regimes.

## 1 Introduction

Approximating functions and solving PDEs plays an important role in scientific computing since many problems in science and engineering can be represented by a set of partial differential equations (PDEs) through mathematical modeling (Huang et al. (2022)). Conventionally, numerical algorithms such as the Finite Element Method (FEM) (Hughes (2000)) and Finite Difference Method (FDM) (Smith (1985)) are widely employed to tackle this task. One drawback of conventional numerical methods is: as the dimensionality of the problem increases (e.g., high-dimensional PDEs), the computational cost of FDM and FEM grows exponentially. Grohs et al. (2023) proved neural networks can overcome this curse of dimensionality in the approximation of Black-Scholes PDEs. While conventional neural networks rely on back-propagation to determine their trainable parameters, Bolager et al. (2023) introduced sampling-based neural network, which accelerated the training of the neural network. Moreover, Jagtap et al. (2020) showed that adaptive trainable parameters in the activation function can bring extra approximation power to the neural network.

Inspired by the aforementioned works, in our work, we propose an innovative method that combines a sampling-based framework with additional trainable parameters in the activation functions. We compared the performance and training time of our approach with the conventional methods. We also discussed the limitations of our method and outlined potential directions for future research.

We provide an overall description of the numerical task to be accomplished in Section 3. We introduce our method in Section 4 to combine sampling-based training with adaptive activation functions. We conduct various numerical experiments in Section 7.3, and conclude our work in Section 6 and 6.2.

## 2 Related Works

Huang et al. (2022) provided a comprehensive overview of how DNNs (like PINNs, neural operators) are used to approximate solutions to PDEs, discussing both theory and applications. Uriarte (2024) surveyed recent methods for using neural networks (MLP-based, residual, Ritz-inspired)

in PDE approximation, highlighting their flexibility and mesh-free advantages. Rosenblatt (1958) first proposed multi-layer perception, which formed the basis architecture of multi-layer feedforward neural networks (MLP). He also applied SGD to optimize his perceptron model. Rumelhart et al. (1986) popularized the back-propagation method, enabling effective training of multi-layer neural networks by iteratively adjusting connection weights to minimize output error. The back-propagation method is also employed in our work as a baseline for comparison against sampling-based approaches. Dubey et al. (2022) provided a comprehensive review of various activation functions used in deep learning. In our work, we employ several fundamental activation functions primarily as baselines for comparison against our proposed approach (with adaptive activation functions). Ramachandran et al. (2017) proposed automatic search techniques to discover new activation functions. Jagtap et al. (2020) introduced the adaptive activation function and suggested that it changes the topology of the loss function involved in the optimization process and has better learning capabilities than the traditional one (fixed activation), as it greatly improves the convergence rate, especially at early training. Boullé et al. (2020) introduced a rational neural network. They proved theoretically that rational neural networks outperform ReLU networks by deriving their error bounds. Molina et al. (2020) studied a special formulation of rational function called the Pade Approximation Unit that is safe to poles and unbounded values when used as an activation function. Bolager et al. (2023) proposed the "Sampling Where it Matters" (SWIM) method to train neural networks. This breakthrough paved the way for an entirely novel approach to determining the weights and biases of a neural network, circumventing the computationally expensive back-propagation method. Datar et al. (2024) used the SWIM method to approximate initial conditions and then iteratively solve the ODE subproblem numerically to obtain the overall solution of the PDE. This work presented an demonstration of the SWIM method's potential for solving PDEs, which is also one of the motivations behind our work.

## 3 BACKGROUND

**Numerical Task**. In this work, the main task we want to accomplish is to use a neural network to solve PDEs. Usually, this is done by letting the neural network learn the true solution $u(\mathbf{x})$ of the PDE, which is typically simulated by numerical algorithms. In practice, the solution $u$ is usually a function that is defined on a region of interest, this region of interest can be a subset of $\mathbb{R}^d$, i.e., $\mathbf{x} \in \mathbb{R}^d$. In our work, we only consider scalar function values. These restrictions are described in the following equations:

$$u(\mathbf{x}) : \mathbb{R}^d \to \mathbb{R} \tag{1}$$

$$\mathbf{x} \in \mathcal{D} \subset \mathbb{R}^d \tag{2}$$

Typically, the data dimension is $d = 2$ and the domain of interest is restricted to $\mathcal{D} = [-20, 20] \times [0, 40]$.

**Ground Truth Dynamic**. A dynamic is defined as a multidimensional mapping, i.e., a function. The dynamic is either explicitly given by a closed analytical form or implicitly expressed as a PDE or ODE that it must satisfy. In real numerical experiments, we constrain the input variable in a region of interest, and we focus on the behavior of the dynamic within this region. The numerical representation of a dynamic is usually expressed as a dataset containing a large amount of discrete input data points and their corresponding function values. The input data points are either grid-based or randomly distributed in the region of interest. In supervised problem settings, we use exactly this dataset to train neural networks.

## 4 METHODS

From the earlier sections, we have the facts:

1. Jagtap et al. (2020) showed that adaptive neural networks have better approximation ability than non-adaptive neural networks. However, we need to perform back-propagation and gradient descent to train the network, which is time-consuming.

2. Bolager et al. (2023) developed SWIM, which determines its network parameters through sampling, which is faster than back-propagation methods.

From these two facts, we try to introduce adaptive SWIM (a-SWIM), which enables the adaptive activation functions (specifically, the rational activation function) to be used in SWIM networks. We expect to modify SWIM by introducing extra trainable adaptive parameters (specifically, the coefficients of the rational activation function) and avoiding back propagation at the same time. We expect to combine the advantages of both the adaptive activation function and the SWIM principle. We now introduce our method step by step.

## 4.1 XU-POINT SETS

We randomly choose $N$ pairs of x-points from the training dataset $\mathbf{X}_{train}$, we denote the chosen x-point pairs as in equation 3

$$(\mathbf{x}_n^{(s)}, \mathbf{x}_n^{(e)})$$
$$n = 1, ..., N$$
(3)

Then for each $(\mathbf{x}_n^{(s)}, \mathbf{x}_n^{(e)})$, we insert $K$ evenly spaced intermediate points along the straight line segment connecting them, as shown in Eq equation 4.

$$\mathbf{x}_n^{(k)} = \mathbf{x}_n^{(s)} + \frac{k}{K+1}(\mathbf{x}_n^{(e)} - \mathbf{x}_n^{(s)})$$
$$k = 1, ..., K$$
(4)

In our work, we set $K = 5$, the $(\mathbf{x}_n^{(s)}, \mathbf{x}_n^{(e)})$ together with the 5 interpolation points are together called x-point sets, denoted in Eq equation 5.

$$(\mathbf{x}_n^{(s)}, \mathbf{x}_n^{(1)} \mathbf{x}_n^{(2)} \mathbf{x}_n^{(3)} \mathbf{x}_n^{(4)} \mathbf{x}_n^{(5)} \mathbf{x}_n^{(e)})$$
(5)

For each element in each x-point set, we extract the corresponding function value from the training dataset $\mathbf{u}_{train}$, note that because of interpolation, not all function values are precisely provided in $\mathbf{u}_{train}$, for those interpolation points whose function values are not present in the dataset, we take the function values of their closest neighboring points, see in equation 6.

$$u_n^{(k)} = \begin{cases} u(\mathbf{x}_n^{(k)}) & \mathbf{x}_n^{(k)} \in \mathbf{X}_{train} \\ u(\underset{\mathbf{x} \in \mathbf{X}_{train}}{\arg \min} ||\mathbf{x} - \mathbf{x}_n^{(k)}||) & \mathbf{x}_n^{(k)} \notin \mathbf{X}_{train} \end{cases}$$
(6)

$$u_n^{(s)}, u_n^{(e)} = u(\mathbf{x}_n^{(s)}), u(\mathbf{x}_n^{(e)})$$
(7)

(8)

The x-point sets in equation 5 and the function values in equation 6 and equation 7 are together called the xu-point sets, denoted in equation 9.

$$(\mathbf{x}_n^{(s)}, \mathbf{x}_n^{(1)}, \mathbf{x}_n^{(2)}, \mathbf{x}_n^{(3)}, \mathbf{x}_n^{(4)}, \mathbf{x}_n^{(5)}, \mathbf{x}_n^{(e)})$$
$$(u_n^{(s)}, u_n^{(1)}, u_n^{(2)}, u_n^{(3)}, u_n^{(4)}, u_n^{(5)}, u_n^{(e)})$$
(9)

Figure 1 visualizes what we did so far, as one can see, the xu-point sets are visualized as red, blue crosses, and pink dots.

## 4.2 WEIGHTS AND BIASES OF a-SWIM

The way we determine the weights and biases of adaptive SWIM is almost identical to the conventional SWIM, as shown in equation 10

$$\mathbf{w}_n = (s_2 - s_1)\frac{\mathbf{x}_n^{(e)} - \mathbf{x}_n^{(s)}}{||\mathbf{x}_n^{(e)} - \mathbf{x}_n^{(s)}||^2} \in \mathbb{R}^d$$
$$b_n = -\mathbf{w}_n^T \mathbf{x}_n^{(s)} + s_1 \in \mathbb{R}$$
(10)

Where $s1$ and $s2$ are constants. Note that these $\mathbf{w}_n$ and $b_n$ are just candidates, not the really applied weights and biases in our network. We design the weights and biases as such, so that the region we intend to approximate (the x-point pair) is precisely mapped to the region $[s1, s2]$. This means that we align the most expressive and numerically stable region of the activation function with the region we intend to approximate, see in equation 11. In our work, $s_1 = -1, s_2 = 1$.

$$\mathbf{w}_n^T \mathbf{x}_n^{(s)} + b_n = s_1$$
$$\mathbf{w}_n^T \mathbf{x}_n^{(e)} + b_n = s_2$$
(11)

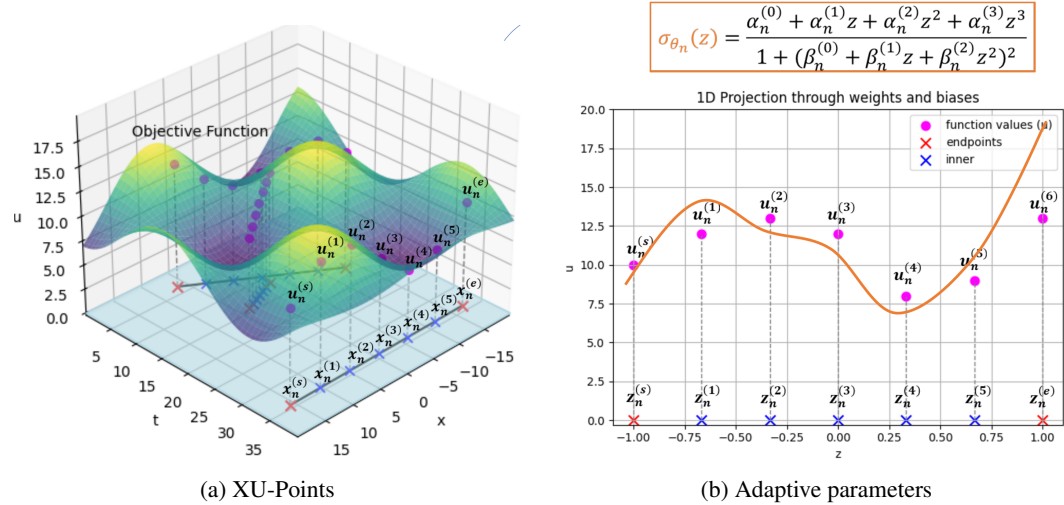

(a) XU-Points           (b) Adaptive parameters

Figure 1: XU-point Sets and Adaptive Parameters

### 4.3 FIND ADAPTIVE PARAMETERS

We use a rational function as the adaptive activation function. Instead of using the original rational function, we adopt a variant of Pade Approximation Unit (PAU) (Molina et al. (2020)). Although the original PAU formulation avoids poles by design, it includes an absolute value term in the denominator, we observed that this structure introduces unstable gradient behavior, especially when the absolute value term approaches zero but remains non-smooth. To address this, we replace the absolute value with a squared term in the denominator, i. e. we redefine the activation as a rational function with a squared polynomial denominator, which guarantees smoothness and avoids gradient discontinuities. To maintain degree consistency between the numerator and denominator and to preserve the expressivity of the function, we also reduced the degree of the denominator accordingly. This modification provides more stable gradients during training while retaining the functional flexibility of the original PAU. This modified version of PAU is written in equation 12.

$$\sigma_{\boldsymbol{\theta}_n}(z) = \frac{\sum_{i=0}^{r_p} \alpha_n^{(i)} z^i}{1 + (\sum_{j=0}^{r_q} \beta_n^{(j)} z^j)^2} \tag{12}$$

$$\boldsymbol{\theta}_n = [\alpha_n^{(0)}, ..., \alpha_n^{(r_p)}, \beta_n^{(0)}, ..., \beta_n^{(r_q)}]$$

Where the adaptive parameters $\boldsymbol{\theta}_n$ denote the coefficients of the numerator and denominator polynomials. By setting $r_p = 3, r_q = 2$, we set the polynomial inside the squared denominator to be of 2nd-order, so that after squaring, the overall denominator becomes a 4th-order polynomial. This ensures that the denominator has a slightly higher order than the cubic numerator, providing both bounded output behavior and sufficient flexibility while maintaining a rational structure. Now our task is to determine $\boldsymbol{\theta}_n$. To do that, we first apply a linear transformation using the candidate weights and biases calculated in equation 10 to map the x-point sets defined in equation 5 into 1-dimensional space, see in equation 13.

$$z_n^{(s)} = \mathbf{w}_n^T \mathbf{x}_n^{(s)} + b_n$$
$$z_n^{(e)} = \mathbf{w}_n^T \mathbf{x}_n^{(e)} + b_n$$
$$z_n^{(k)} = \mathbf{w}_n^T \mathbf{x}_n^{(k)} + b_n \tag{13}$$
$$k = 1, ..., K$$

Because the $\mathbf{w}_n$ and $b_n$ are strategically designed, the 1-d projections $(z_n^{(s)}, z_n^{(1)}, z_n^{(2)}, z_n^{(3)}, z_n^{(4)}, z_n^{(5)}, z_n^{(e)})$ are uniformly distributed between -1 and 1, as shown in Figure 1. $\boldsymbol{\theta}_n$ is then obtained by a sub-optimization problem minimizing some loss functions. We offer in our work two alternative loss functions: a)The sub-optimization problem using the MSE loss function is defined in equation 14. b)The sub-optimization problem using the cosine loss

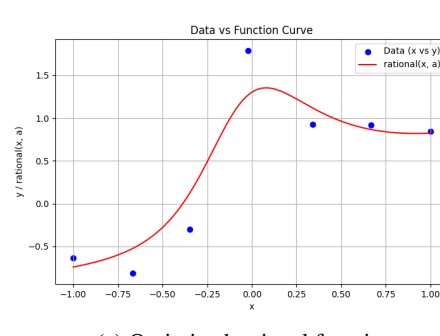 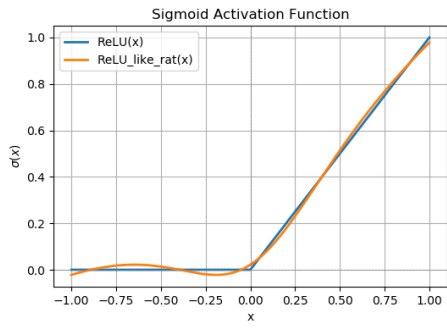

(a) Optimized rational function    (b) Initialization of adaptive parameters

Figure 2: Initialization and optimization of rational activation function

function is defined in equation 15.

$$\boldsymbol{\theta}_n^* = \arg\min_{\boldsymbol{\theta}_n} E_{se}(\boldsymbol{\theta}_n) + \sum_{k=1}^{K} (\sigma_{\boldsymbol{\theta}_n}(z_n^{(k)}) - u_n^{(k)})^2 + \lambda||\boldsymbol{\theta}_n||_2^2 \tag{14}$$

$$E_{se}(\boldsymbol{\theta}_n) = (\sigma_{\boldsymbol{\theta}_n}(z_n^{(s)}) - u_n^{(s)})^2 + (\sigma_{\boldsymbol{\theta}_n}(z_n^{(e)}) - u_n^{(e)})^2$$

$$\boldsymbol{\theta}_n^* = \arg\min_{\boldsymbol{\theta}_n} (1 - cos^2(\boldsymbol{\theta}_n)) + \lambda||\boldsymbol{\theta}_n||_2^2$$

$$cos^2(\boldsymbol{\theta}_n) = \frac{(\sigma_{\boldsymbol{\theta}_n}(z_n^{(s)})u_n^{(s)} + \sigma_{\boldsymbol{\theta}_n}(z_n^{(e)})u_n^{(e)} + \sum_{k=1}^{K}\sigma_{\boldsymbol{\theta}_n}(z_n^{(k)})u_n^{(k)})^2}{\sigma_{\boldsymbol{\theta}_n}^2(z_n^{(s)}) + \sigma_{\boldsymbol{\theta}_n}^2(z_n^{(e)}) + \sum_{k=1}^{K}\sigma_{\boldsymbol{\theta}_n}^2(z_n^{(k)})} \tag{15}$$

Where $\lambda$ is the L2-regularization factor, and this is set in our work to $\lambda = 10^{-6}$. We design the loss function based on cosine similarity to encourage strong alignment between the fitted function and the scattered data points. In addition to the choice of loss function, there are also multiple options for initializing the trainable adaptive parameters. We consider three initialization methods, which are described in detail below: a)We initialize all adaptive activation functions, here in our case, a rational function, with a specific set of adaptive parameters such that it behaves like ReLU in the region of [-1,1], see in equation 16. We plot this ReLU-like initialization in Figure 2 right. b)Alternatively, we can also randomly initialize the adaptive parameters between 0 and 2. c)We initialize all adaptive parameters except the 0-power term in the numerator as zero. In practice, numerical experiment showed that this initialization is unstable, so we exclude this method in our final work.

$$\sigma(x) = \frac{0.0218 + 0.5x + 1.5957x^2 + 1.1915x^3}{1 + 2.383x^2} \tag{16}$$

With both the initialization method and loss function specified, we proceed to the optimization phase. Figure 2 left visualizes the sub-optimization problem, we are trying to find the optimal $\boldsymbol{\theta}_n$ such that the activation function (the orange curve) fits the scatter points defined by $(z_n^{(k)}, u_n^{(k)})$ as better as possible. In practice, we employ the Adam optimizer (Kingma & Ba (2014)) due to its stability and efficiency in handling adaptive parameters. Because all these optimization problems are independent of each other, they can be easily parallelized. Since each individual optimization problem only involves few trainable parameters, solving these problems parallel is computationally more efficient than optimizing large trunk of parameters iteratively (what we do in back-propagation based neural networks).

### 4.4 PROBABILITY

At this point, we have all the necessary information $(\mathbf{w}_n, b_n, \boldsymbol{\theta}_n)$ to construct a complete neuron. Based on the preceding calculations and methodology, each neuron models only a small, localized

subset of the dataset (i.e., the xu-point set), thereby providing a limited and focused view of the overall data distribution. Since the initial candidate pool contains a large number of neurons ($N$), far exceeding the number we ultimately need ($M$), it becomes essential to identify which neurons are most valuable. In this section, we define a probability distribution over the $N$ candidate neurons, reflecting their relative importance or relevance. We then sample $M$ neurons from this distribution, thereby selecting a compact yet representative subset for further modeling. We present 3 alternatives for calculating the probability distribution.

From preceding methodologies, we know that each neuron corresponds to a xu-point set, which it tries to model. We take the variance of the $u$-values of the xu-point set as the probability assigned to its corresponding neuron. Our intuition is that when the $u$-values within a region deviate significantly from their mean, the underlying function exhibits complex or highly variable behavior in that area. Consequently, a neuron responsible for modeling such a region is considered more valuable—regardless of its actual approximation accuracy—since it targets a structurally informative subset of the input space. This type of probability is defined in equation 17.

$$\mathbf{p}_1, \quad p_{1,n} \propto \text{var}(u_n^{(s)}, u_n^{(1)}, u_n^{(2)}, u_n^{(3)}, u_n^{(4)}, u_n^{(5)}, u_n^{(e)}) \tag{17}$$

Although the variance-based probability enables efficient computation, it is too intuitive and straightforward. We introduce here a more principled and interpretable alternative to compute the probability–the cosine-based probability. We first plug in the whole training set $\mathbf{X}_{train}$ into each neuron, and we obtain the $\mathbf{F}$ matrix, as shown in equation 18.

$$\boldsymbol{\sigma}_n = \sigma_{\boldsymbol{\theta}_n}(\mathbf{X}_{train}^T \mathbf{w}_n + b_n \mathbf{1}) \in \mathbb{R}^T$$
$$\mathbf{F} = [\boldsymbol{\sigma}_1, ..., \boldsymbol{\sigma}_N] \in \mathbb{R}^{T \times N} \tag{18}$$

$\mathbf{F}$ forms a basis of its column space. Each candidate neuron corresponds to one column of $\mathbf{F}$. We need to select $M$ columns out of $\mathbf{F}$ as the neurons in our network. To formalize this selection, we define a column selection matrix $\mathbf{S} \in \mathbb{R}^{N \times M}$, where $\mathbf{FS} \in \mathbb{R}^{T \times M}$ represents the $M$ chosen columns. Ideally, the selected columns should maximize the alignment with the objective function. Specifically, the optimal selection is the matrix $\mathbf{S}$ that maximizes the projection energy of the ground truth $\mathbf{u}_{train}$ onto the column space spanned by selected columns $\mathbf{FS}$. We formalize this selection in equation 19.

$$\mathbf{S}^* = \arg\max_{\mathbf{S}}(\mathbf{u}_{train}^T \mathbf{FS}(\mathbf{S}^T \mathbf{F}^T \mathbf{FS})^{-1} \mathbf{S}^T \mathbf{F}^T \mathbf{u}_{train}^T) \tag{19}$$

It is not easy to get the exact solution of $\mathbf{S}^*$. But we can introduce some assumptions to get rough guidance toward the true solution. Since $\mathbf{S}$ is column selection matrix, $\mathbf{S}^T \mathbf{S} = \mathbf{I}$. If $\mathbf{F}^T \mathbf{F} = \boldsymbol{\Lambda}$, where $\boldsymbol{\Lambda}$ is diagonal matrix, then equation 19 collapse to:

$$\mathbf{S}^* = \arg\max_{\mathbf{S}}(\mathbf{u}_{train}^T \mathbf{F} \boldsymbol{\Lambda}^{-\frac{1}{2}} \mathbf{SS}^T \boldsymbol{\Lambda}^{-\frac{1}{2}} \mathbf{F}^T \mathbf{u}_{train}) \tag{20}$$

$$= \arg\max_{\mathbf{S}} ||\mathbf{S}^T \boldsymbol{\Lambda}^{-\frac{1}{2}} \mathbf{F}^T \mathbf{u}_{train}||^2 \tag{21}$$

One can interpret equation 21 as we select those $M$ columns of $\mathbf{F}$ with the highest squared cosine value with respect to $\mathbf{u}_{train}$, this is because $\boldsymbol{\Lambda}^{-\frac{1}{2}}$ stores all inverse l2-norms of $\mathbf{F}$ columns, and the cosine value is exactly given by inner-product divided by l2-norm. Based on this observation, we take the squared cosine value as the probability, as shown in equation 22.

$$\mathbf{p}_2, \quad p_{2,n} \propto \frac{(\boldsymbol{\sigma}_n^T \mathbf{u}_{train})^2}{\boldsymbol{\sigma}_n^T \boldsymbol{\sigma}_n} \tag{22}$$

Another alternative to compute probability arises when we follow the steps in the previous paragraph and impose $\mathbf{F}^T \mathbf{F} = \boldsymbol{\Lambda} = \mathbf{I}$, then equation 21 further collapses into equation 23.

$$\mathbf{S}^* = \arg\max_{\mathbf{S}} ||\mathbf{S}^T \mathbf{F}^T \mathbf{u}_{train}||^2$$
$$= \arg\max_{\mathbf{S}} ||\mathbf{S}^T (\mathbf{F}^T \mathbf{F})^{-1} \mathbf{F}^T \mathbf{u}_{train}|| \tag{23}$$

This means we should select those $M$ $\mathbf{F}$ columns with the highest squared projection coefficients w.r.t. $\mathbf{u}_{train}$, this kind of probability is defined in equation 24.

$$\mathbf{p}_3, \quad p_{3,n} \propto \alpha_n^2, \quad \boldsymbol{\alpha} = (\mathbf{F}^T \mathbf{F})^{-1} \mathbf{F}^T \mathbf{u}_{train} \tag{24}$$

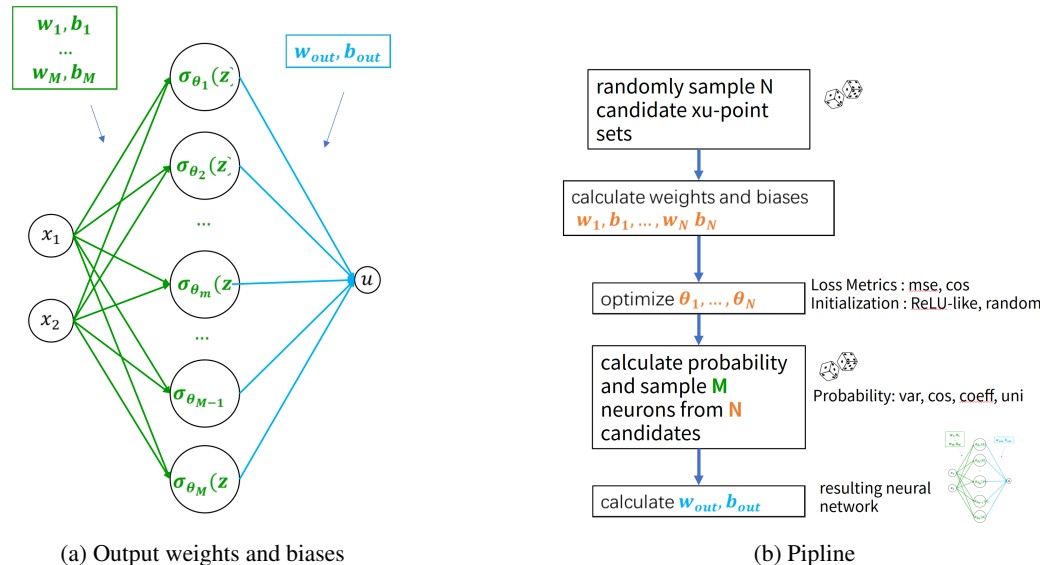

(a) Output weights and biases (b) Pipline

Figure 3: Output weights and pipeline

In reality, neither condition $\mathbf{F}^T\mathbf{F} = \mathbf{\Lambda}$ nor $\mathbf{\Lambda} = \mathbf{I}$ is true. Neither cosine nor coefficient-based probability gives us the exact optimal selection, therefore, we use this probability as guidance for a random approach, not a hard constraint.

We then sample $M$ neurons from the $N$ candidates with respect to the probability distributions, either equation 17, equation 22, or equation 24. We therefore determined all weights, biases, and adaptive parameters of the adaptive SWIM network, see in green terms in Figure 3 left.

### 4.5 OUTPUT WEIGHTS AND BIASES

Now we would like to compute the output weights and biases (blue terms in Figure 3 left) of adaptive SWIM. We denote the $M$ selected $\mathbf{F}$-columns (neurons) in last step as $\mathbf{F}_M$, then the outer weights and biases is given by equation 25:

$$\mathbf{w}_{out}, b_{out} = \arg\min_{\mathbf{w},b} ||\mathbf{u}_{train} - \mathbf{F}_M\mathbf{w} - b \cdot \mathbf{1}||_2 \tag{25}$$

This can be solved by a simple SVD or least squares problem. Til this point, we finished the training of adaptive SWIM, we summarize the whole training pipeline of adaptive SWIM in Figure 3 right. As one can see from the pipeline, the alternative options are listed close to the right of the respective modules, the dices mean that the respective module involves at least one random approach.

### 4.6 INFERENCE OF A-SWIM

Having completed training, we now perform inference on the entire dataset $\mathbf{X}$, which includes both the training set $\mathbf{X}_{trian}$ and the validation set $\mathbf{X}_{val}$. The predicted function value $\hat{\mathbf{u}}$ is given by equation 26.

$$\boldsymbol{\sigma}_m = \sigma_{\boldsymbol{\theta}_m}(\mathbf{X}^T\mathbf{w}_m + b_m\mathbf{1}) \in \mathbb{R}^C$$
$$\mathbf{F}_M = [\boldsymbol{\sigma}_1, ..., \boldsymbol{\sigma}_M] \in \mathbb{R}^{C \times M} \tag{26}$$
$$\hat{\mathbf{u}} = \mathbf{F}_M\mathbf{w}_{out} + b_{out}\mathbf{1} \in \mathbb{R}^C$$

We use the average mean squared error (MSE) to evaluate the performance of the adaptive SWIM network, the average MSE is given by equation 27.

$$MSE = \frac{1}{C}||\mathbf{u} - \hat{\mathbf{u}}||_2^2 \tag{27}$$

Where $\hat{\mathbf{u}}$ is the predicted function value. The smaller the MSE value, the better is the performance of the network.

## 5 OBSERVATIONS

To systematically evaluate the performance and properties of the proposed adaptive SWIM (a-SWIM) network, we conduct a series of numerical experiments designed to highlight the strengths and potential trade-offs of adaptive SWIM. To maintain a clear and focused storyline, we organize our experimental observations into two parts: **a-SWIM vs. SWIM**, (Section 5.1), **a-SWIM vs. BP-NN**, (Section 5.2). All our experiments are conducted on an NVIDIA RTX3080 GPU. The experimental setup, including dataset descriptions, neural network architecture, is provided in detail in Section 7.2.2.

### 5.1 A-SWIM VS. SWIM

In this section, we focus on the comparison between adaptive SWIM (a-SWIM) and its non-adaptive counterpart (SWIM). The comparison includes 3 metrics: prediction accuracy, training time, and the predicted dynamics across the region of interest. The goal of this comparison is to assess the benefits introduced by adaptive parameterization (rational activation function). Since a-SWIM includes multiple design choices, unless otherwise specified, we always fix a-SWIM's design choice to the default setting in order to enable a fair comparison with the standard SWIM model. The default design choice for a-SWIM is:

1. **Probability Strategy**: variance based, see 17.
2. **Repetition Factor**: $r = 2$
3. **Loss Metric**: MSE, see 14.
4. **Activation Function**: rational activation function
5. **Initialization**: ReLU-like

#### 5.1.1 ACCURACY & TRAINING TIME

We compare the MSE loss that can be reached by adaptive SWIM (a-SWIM) and non-adaptive SWIM networks across different network widths and different objective functions, see in Experiment 1. We conclude that our method (a-SWIM) outperformed all conventional SWIM methods under at least 3 of 6 in total testing objective functions, and it also maintained good performance (rank 2) when approximating the remaining 3 objective functions, proving its robustness and versatility. We also compared the required training time for adaptive and non-adaptive SWIM networks in Experiment 2. One can see from this experiment that a-SWIM required roughly twice the training time compared to SWIM. Despite requiring twice the training time relative to SWIM, a-SWIM remains in the order of seconds. This is substantially faster than BP-NNs, which typically take minutes to converge.

#### 5.1.2 PREDICTED DYNAMICS

In this section, we present the predicted dynamics provided by trained a-SWIM vs. SWIM. One can see from Experiment 5 that a-SWIM achieved better accuracy compared to SWIM of the same width. This is the case in the "KdV_sine", "Euler-Bernoulli" and "Advection" objective functions. Notably, SWIM can also match the accuracy of a-SWIM, but at the cost of a larger network width. Let us examine the case with the largest network width, where both a-SWIM and SWIM achieve high overall accuracy. While the quantitative performance is similar, the error map of a-SWIM is noticeably cleaner, exhibiting fewer high-error regions. In contrast, the error map of the SWIM network contains distinct bright spots, indicating multiple localized errored-areas. We also found from Experiment 6 that for both a-SWIM and SWIM, the jumping edge of the objective function is always the most difficult part to approximate, because there is always a bright curve remaining in the error maps of both networks.

To summarize this section, we showed through experiments that the a-SWIM achieved better accuracy in 3/6 cases, and is never the worst one among all SWIM alternatives, see Exp. 5. Both a-SWIM and SWIM fit the low-frequency parts first and then the high-frequency parts, see Exp. 6. A-SWIM requires roughly twice as much training time compared to SWIM, see Experiment 2. When we decide to use a-SWIM, we are trading a few seconds of extra training time for either better accuracy (same network width) or smaller network width (same level of approximation accuracy).

## 5.2 A-SWIM VS. BP-NN

In this section, we focus on the comparison of a-SWIM vs. BP-NN. We aim to quantify the reduction in training time achieved by removing backpropagation, as well as the corresponding impact on model accuracy. We compare the MSE loss that can be reached by a-SWIM vs. BP-NN. Note that BP-NNs are not limited to single-layer architectures. Therefore, using network width as a basis for comparison is not meaningful. Instead, we compare the models based on their total number of trainable parameters, which provides a more consistent benchmark, see in Experiment 3. We conclude that while a-SWIM generally performs worse than BP-NNs in terms of accuracy, the performance gap remains limited: For models with a large number of parameters, the MSE achieved by a-SWIM is at most one order of magnitude worse than that of BP-NN. We also compare the required training time of a-SWIM vs. BP-NN, where we set the maximum iteration steps of the latter to 10000. We can conclude from Experiment 4 that a-SWIM requires significantly less time to be trained. In our experiment, a-SWIM is at least 20 times faster than BP-NNs.

## 6 CONCLUSIONS & OUTLOOKS

### 6.1 PERFORMANCE EVALUATION & LIMITATIONS

Inspired by adaptive activation function used in neural networks and sampling based training approach, we proposed our innovative computation framework to combine adaptive activation function with sampling approach, we aim to improve the approximation capability while preserving the training speed advantage brought by sampling.

After developing a-SWIM and conducting various numerical experiments to compare a-SWIM vs. other approaches, we confirm that a-SWIM outperforms conventional SWIM in 3 out of 6 cases (Exp. 1). A-SWIM typically trades a slightly longer training time for either improved accuracy at the same network width, or reduced width while maintaining similar accuracy (Exp. 2, 6, 5, 7). A-SWIM preserved the key advantage of sampling-based neural networks: it's orders of magnitude faster than backpropagation-based methods (Exp. 4).

The key drawbacks of a-SWIM we found in our experiment are listed as follows: 1) One must take care of the rational function parametrization carefully, making sure there are no poles that lead to infinite values. 2) Both a-SWIM and conventional SWIM performed poorly when approximating objective functions with extremely high-frequency parts (especially when the frequency is comparable to the data resolution of the input region). 3) Although a-SWIM typically outperforms SWIM at the same network width, it involves a significantly larger number of parameters. However, this issue is expected to diminish in the multi-layer setting, where the relative contribution of the additional adaptive parameters becomes negligible compared to the total number of weights and biases.

### 6.2 OUTLOOKS

**Multi-Layered a-SWIM**. While Bolager et al. (2023) considered a multi-layer SWIM network in their work, we currently only considered the one-layer case and its adaptive version. We expect a multi-layer a-SWIM network to extract deeper data features as it does in back-propagation-based neural networks.

**Other Activation Functions**. The concept of adaptive activation function covers a wide range of functions, we in our work only considered rational activation function and rather, its no-poles special formulation, because it's clear parametrized and computationally efficient. It's compromising to take more activation functions into account and study which form of activation functions is suitable for which type of objective functions.

**Solution To PDEs**. In our work, we focused on solving PDEs by directly approximating their solution (usually simulated by numerical algorithms). We may also solve the PDEs directly by approximating the right-hand side of the PDE with a-SWIM, and then perform operations on its activation functions (which is easy, since they are clear parametrized rational functions) to retrieve the true solution of the PDE.

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

# 7 APPENDIX

## 7.1 OBJECTIVE FUNCTIONS

Objective functions are the goal dynamics we would like to approximate using neural networks. In our experiments, we consider $f : \mathbb{R}^2 \to \mathbb{R}$ as objective functions. They are all real-valued functions defined on a 2d region of interest $x \in [-20, 20], t \in [0, 40]$.

**KdV_sine**. The analytical form of KdV_sine PDE is given in equation 28, we set the initial condition as in equation 29:

$$u_t = -uu_x - u_{xxx} \tag{28}$$
$$u(x, 0) = -\sin(\pi x/20) \tag{29}$$

There is no analytical form of the solution of the PDE in equation 28, so we simulate the solution $u$ numerically.

**Discontinuous Complicated**. In order to evaluate the model performance under the objective function with high frequency and jumping edge, we designed the objective function named as "discontinuous complicated" and "discontinuous trivial". Both of these functions contain high frequency parts and discontinuity.See in equation 30

$$u(x, t) = \begin{cases} sin(6q) & \text{if} q < 0 \\ 5 + 0.5qcos(6q) & \text{if} q \geq 0 \end{cases} \tag{30}$$

$$q = sin(0.1x) + cos(0.05t) - 0.5sin(0.02xt) \tag{31}$$

The locus $q$ in equation 31 defined a curved boarder across the region of interest, dividing the region of interest into 2 areas, within each area, the objective function exhibits completely different behavior and hence shows obvious discontinuity. One can easily see the discontinuity between the two sides of locus $q$ in the corresponding subfigure in Figure 4. On both sides of the locus, high-frequency parts are present.

**Advection.** The advection equation is a fundamental partial differential equation that describes the transport of a physical quantity (such as heat, mass, or concentration) through a medium by the motion of that medium itself (Wikipedia contributors (2024)). The advection partial differential equation is defined in equation 32:

$$u_t + \beta u_x = \nu u_{xx} \tag{32}$$

Where $\beta$ is constant, $\nu$ is the kinematic viscosity. In our work, we set $\beta = 8$ and $\nu = 0$, the advection PDE becomes equation 33. We solve this PDE with the initial condition defined in equation 34, and the closed form solution can be written as equation 35.

$$u_t + 8u_x = 0 \tag{33}$$

$$u(x, 0) = \sin(\frac{\pi}{20}x) \tag{34}$$

$$u(x, t) = \sin(\frac{\pi}{20}x - \frac{2\pi}{5}t) \tag{35}$$

**Burgers.** The Burgers equation, particularly in its inviscid form, is a canonical model for studying wave steepening and shock formation (Burgers (1948)). The burgers equation, the initial condition, and the boundary condition are written in equation 36, equation 37, equation 38, and equation 39. Although we don't have the closed-form solution of the Burgers equation, we get the training data by simulating the Burgers equation numerically.

$$u_t + uu_x = \nu u_{xx} \tag{36}$$

$$u(x, 0) = 5\sin(\frac{\pi}{20}x) \tag{37}$$

$$u(-20, t) = 0 \tag{38}$$

$$u(20, t) = 0 \tag{39}$$

**Discontinuous Trivial**. Similar as "Discontinuous Complicated", we also manually constructed the function "Discontinuous Trivial", it also contains both discontinuity and high-frequency parts, the only difference is that the locus where the jumping takes place is now a straight line.

**Euler Bernoulli**. The Euler–Bernoulli beam equation is a classical partial differential equation that models the transverse deflection of slender beams under bending. The Euler-Bernoulli PDE used in our work is written in equation 40. Where the $f$ represents an external, time-dependent force that is applied to the beam.

$$u_{tt} + u_{xxxx} = f(x, t) \tag{40}$$

To simplify the visualization and make the solution compatible with the region of interest ($x \in [-20, 20], t \in [0, 40]$), we set the external force term $f$ as in equation 41

$$f(x, t) = 5((\frac{\pi}{40})^4 - (\frac{\pi}{10})^2) \cos(\frac{\pi}{10}t) \cos(\frac{\pi}{40}x) \tag{41}$$

For simplicity, we use the initial and boundary conditions defined in equation 42 and equation 43.

$$u(x, 0) = 5 \cos(\frac{\pi}{40}x) \tag{42}$$

$$u_t(x, 0) = 0 \tag{43}$$

The closed form solution of Euler Bernoulli PDE with conditions in equation 41 equation 42 and equation 43 is written in Eq equation 44.

$$u(x, t) = 5 \cos(\frac{\pi}{40}x) \cos(\frac{\pi}{10}t) \tag{44}$$

All the dynamics of the aforementioned objective functions are plotted in Figure 4.

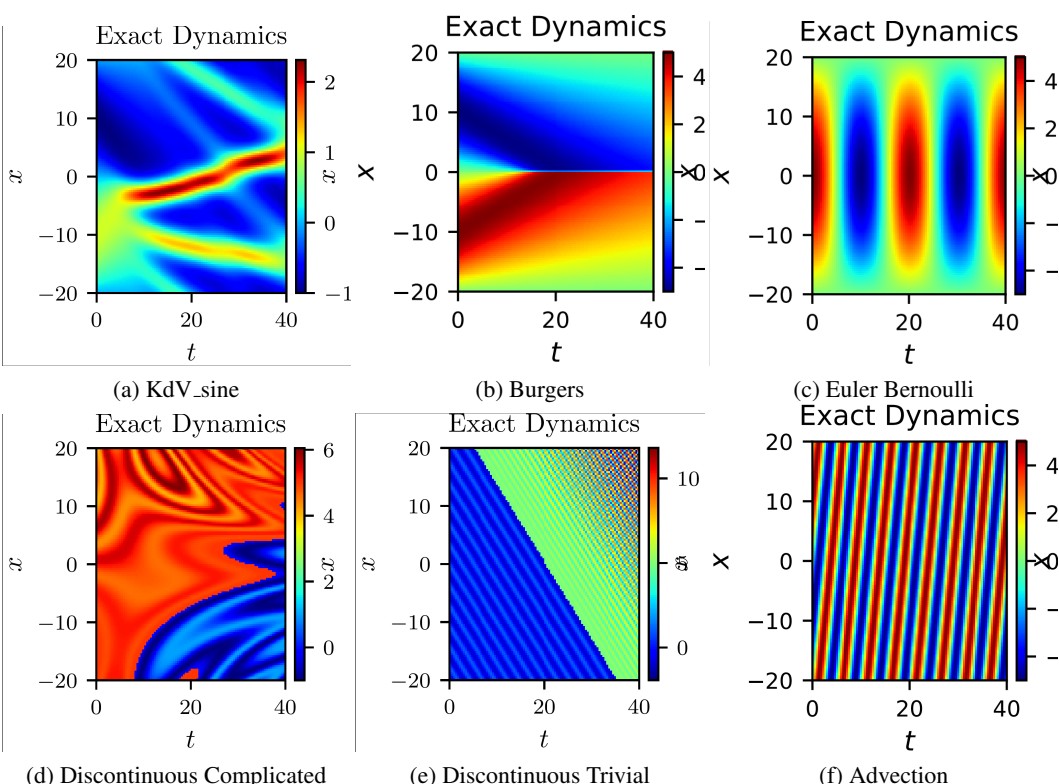

Figure 4: Ground truth objective functions

## 7.2 EXPERIMENT SETTINGS

In this section, we provide all numerical experiments conducted in our work. All numerical experiments in this work were conducted on an NVIDIA GeForce RTX 3080 GPU. The source code link:

### 7.2.1 EXPERIMENT SETTINGS OF BACK-PROPAGATION NETWORKS

In the following experiments, we train different neural networks based on gradient descent and back propagation. To make the experiment results more reliable, we involve different experiment settings.

**Parameter Amounts**. To compare the performance of the BP-NNs with different amounts of parameters, we predefined 3 amount levels, namely A, B, and C. Because we would like to later compare the back propagation performance vs. the SWIM performance, knowing that SWIM networks always give better performance when the network width is large, we defined the parameter amount levels in such a way that the parameter number coincides with the parameter number of the widest 3 SWIM networks, namely 3200, 6400, and 12800.

**Network Shapes**. The depth of the neural networks enables them to extract hierarchical and high-level features from data. We also include this effect in our experiment. To compare different neural networks with different depths, we predefined networks with depths 1,5,10, and we refer to them as "s" for shallow, "m" for medium, and "d" for deep, accordingly. To ensure consistency with respect to parameter amounts we defined previously, the network width is calculated in the following way:

$$P = (L-1)M^2 + (L+3)M + 1 \tag{45}$$

Where $P$ stands for the parameter amounts in the previous section, $M$ stands for the width, and $L$ stands for the depth of neural networks.

To summarize this part, we list the experiment settings of BP-NN as following:

1. **Parameter Amounts**: A:3200, B:6400, C:12800

2. **Network Shapes**: number of hidden layers s:1, m:5, d:10

### 7.2.2 EXPERIMENT SETTINGS OF SWIM

**Network Widths & Repetition Factor**. In this section, we involve the experiment settings of sampling-based neural networks. The objective functions remain the same as in back-propagation-based experiments. We use 1-hidden-layer sampling-based neural networks. The network widths $M$ are set to be 100, 200, 400, 800, 1600, 3200. Since we only focus on 1-layer networks, width corresponds to the number of neurons and hence the number of activation functions. The repetition scaler $r$ is given by the ratio of the pool size $N$ and the network width $M$.

$$r = \frac{N}{M} \tag{46}$$

In other words, we are sampling $M$ instances from an $N$-sized pool. We expect a better result when $r$ is larger, since we have more choices than in a small pool.

**Probability & Optimization Strategies**. For the probability strategies, we deploy 3 different kinds of them. The probabilities are given in equation 17, equation 22, and equation 24, which rely on the variance of $\mathbf{u}$ values, the cosine similarity of $\mathbf{u}$ and the columns of intermediate outputs $\mathbf{F}$, and the coefficients of $\mathbf{u}$ represented in the basis $\mathbf{F}$, respectively. We denote these 3 probability strategies as "var", "cos", and "coeff", respectively. As mentioned in Section 4.3, we obtain the adaptive parameters through the sub-optimization problem given by equation 14. In equation 14, we intend to optimize the MSE loss at the 6 points; alternatively, we can also optimize the cosine similarity at the 6 points. The losses of sub-optimization are denoted as "mse" and "cos" respectively.

**Initialization Methods**. As mentioned in Section 4.3, we have different methods to initialize the adaptive parameters we intend to train, namely relu_like or random initialization.

In this work, we don't compare within these design choices, rather, we use default settings for a-SWIM, which is introduced in Section 5.1.

## 7.3 EXPERIMENTS

**Experiment 1** *In this experiment, we study the accuracy achieved by a-SWIM vs. SWIM at different network widths. We compare the MSE loss of a-SWIM vs. SWIM under different layer widths $M$. Because a-SWIM involves a variety of design choices, it is impractical to present results for*

*all configurations. Therefore, unless otherwise specified, references to a-SWIM in the following refer to its default design choice. The MSE loss curves are presented in Figure 5, each sub-figure represents one objective function, see the corresponding caption. For each sub-figure, the x-axis represents the network width $M$, and the vertical axis represents the MSE loss. The blue, orange, red, and green curves represent a-SWIM, ReLU-SWIM, Sigmoid-SWIM, and Tanh-SWIM, respectively. One can see that the a-SWIM outperforms all other SWIMs in approximating "KdV_sine", "Euler Bernoulli", and "Advection" objective functions. When approximating the "Discontinuous Complicated" function, the a-SWIM performs as good as ReLU-SWIM. While other activation functions each have their strengths, a-SWIM demonstrates consistently strong performance across all objective functions. It may not always be the best in every case, but it remains among the top performers, highlighting its robustness and versatility. The experiment statistics can be found in Table 1. We can conclude from this experiment that a-SWIM outperforms all other SWIMs in 3 out of 6 cases and is never the worst one.*

**Experiment 2** *In this experiment, we examine the required training time of a-SWIM vs. SWIM. The required training time for a-SWIM vs. SWIM can be found in Figure 6. Each sub-figure stands for one objective function, as seen in the caption. For each sub-figure, the x-axis represents the network widths $M$, and the y-axis represents the required training time in seconds. As one can see, while a-SWIM requires roughly twice the training time compared to other SWIMs, it delivers superior and more robust performance across various tasks, see in Experiment 1. The extra time required for training is reasonable because we have to solve multiple sub-optimization problems to calculate the adaptive parameters. We also noticed that as $M$ increases, the training time of each respective network remains nearly constant. This suggests that the training complexity is not significantly affected by the network width, indicating efficient parallelism or scalability in the training process. The experiment statistics are collected in Table 2a. Since the training time for different objective functions is quite similar, we only take the KdV objective function as an example. We can conclude from this experiment that a-SWIM needs about twice as much time to train compared to other SWIMs, and the training time is almost not affected by the growing network widths.*

**Experiment 3** *In this experiment, we compare the best MSE loss reached by a-SWIM and BP-NNs. We set the activation function of BP-NN to a rational function to seek a fair comparison. Since BP-NNs are not necessarily single-layered, it's nonsense measuring them in "network widths", instead, we use the parameter amount. The experiment results are plotted in Figure 7. The x-axis stands for parameter amount, and the y-axis represents the MSE, the blue region represents the best MSE loss achieved by BP-NNs with different parameter amounts as well as network shapes, while the black curve represents the MSE curve of a-SWIM. One can conclude that a-SWIM completely outperforms BP-NN in 1 case, completely underperforms BP-NN in 2 cases, and in the rest cases there are intersections.*

**Experiment 4** *In this experiment, we compare the training time of a-SWIM vs. BP-NN, we use the adaptive Tanh as the activation function of the BP-NN. Since the training time for different objective functions is almost the same, we take the "KdV_sine" as an example, see in Figure 7. Again, the x-axis represents the parameter amount, the y-axis represents the training time in seconds, and the blue regions represent the required training time of differently shaped BP-NNs at that parameter amount level. The experiment statistics are collected in Table 2b. One can see that the training time of a-SWIM is significantly shorter than that of BP-NNs (about 14 seconds vs. 800 seconds)*

**Experiment 5** *In this experiment, we compare the predicted dynamics of a-SWIM vs. SWIM. Figure 8 collects the predicted dynamics of a-SWIM vs. SWIM when approximating objective functions "KdV_sine" (top 4 rows) and "Euler Bernoulli" (bottom 4 rows).*

*Within each 4-rowed part, the first row represents the a-SWIM's predicted dynamics under different layer width $M$, the second row represents the SWIM predicted dynamics, the third row the error maps of a-SWIM, and the fourth row the error maps of SWIM.*

*Note that here we use Tanh-SWIM, since according to Experiment 1, Tanh-SWIM provides the best performance except for a-SWIM by these two objective functions. One can see from the dynamics that a-SWIM converges more accurately than Tanh-SWIM, especially in earlier stages where the network widths are small.*

*Besides, one can easily see the performance gap from the error maps at a width of 1600, where both networks reach their top performance. The scale bar on the right-hand side of the figure showed that the a-SWIM reached a lower error than SWIM (0.015 vs 0.10). Moreover, a-SWIM resolves the error over the whole region more evenly, in contrast, Tanh-SWIM remains multiple significant error spots (the bright spots) at some specific locations. One can also see the same effect by the objective function "Advection" (see Figure 9, top 4 rows). The a-SWIM achieved not only lower error at the same network width but also a smoother error distribution.*

**Experiment 6** *Figure 9 (bottom 4 rows) shows the predicted dynamics of the objective function "Discontinuous Complicated" given by a-SWIM vs. SWIM. We can conclude that SWIM networks also fit low frequency part first, then high frequency parts. The a-SWIM and SWIM (in this experiment, ReLU-SWIM) performed as well as each other, and significantly better than other SWIMs (Tanh-SWIM, Sigmoid-SWIM). We found that for both a-SWIM and SWIM, the jumping edge of the objective function is always the most difficult part to approximate, there is always a bright curve remaining in the error maps of both networks, which shows that the networks were unable to approximate the discontinuity perfectly.*

**Experiment 7** *In this experiment, we compare the predicted dynamics of the objective functions "Discontinuous Trivial" and "Burgers" provided by a-SWIM vs. SWIM. As one can see from Fig 10, SWIM (in this experiment also ReLU-SWIM) performed better than a-SWIM, however, other SWIM (Tanh-SWIM and Sigmoid-SWIM) give even worse performance, see in Fig 5. This tells us that a-SWIM is still competitive among all SWIM networks. Although ReLU-SWIM performed better than a-SWIM when approximating "Discontinuous Trivial" functions, it still failed to converge. We can conclude that all SWIM networks are prone to this discontinuous but high-frequency objective function. One can see from Fig 10 (last two rows) that a-SWIM can resolve error in smooth area better than ReLU-SWIM, as the bright line in the error map only occurs in sharp edges of the objective function, and therefore have only half length comparing to the bright line of ReLU-SWIM's error map. In contrast, ReLU-SWIM may process smooth areas worse than a-SWIM, but it can process sharp edges more accurately, as the bright line in the error map is thinner than that in a-SWIM (the last row).*

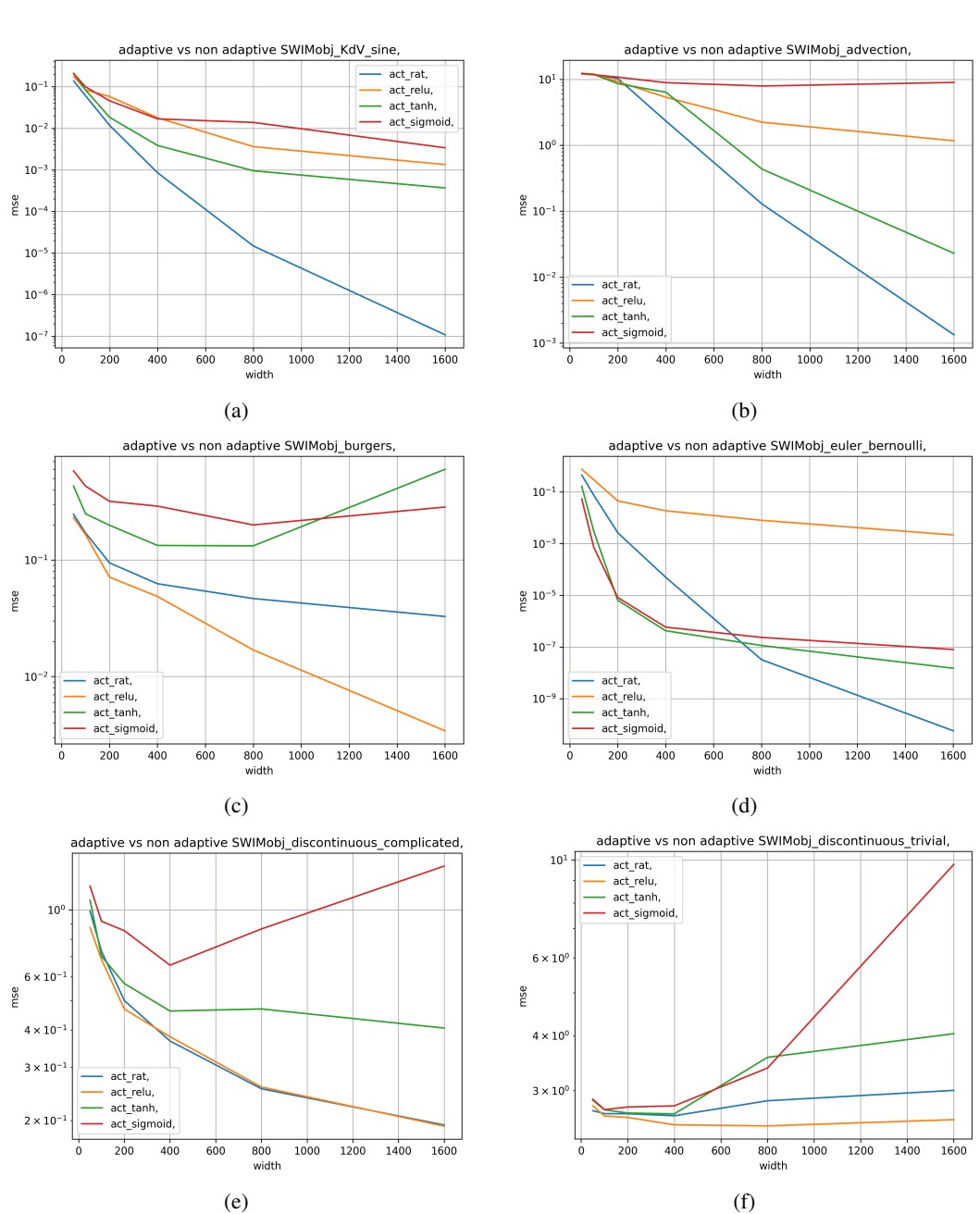

Figure 5: MSE loss a-SWIM vs. SWIM

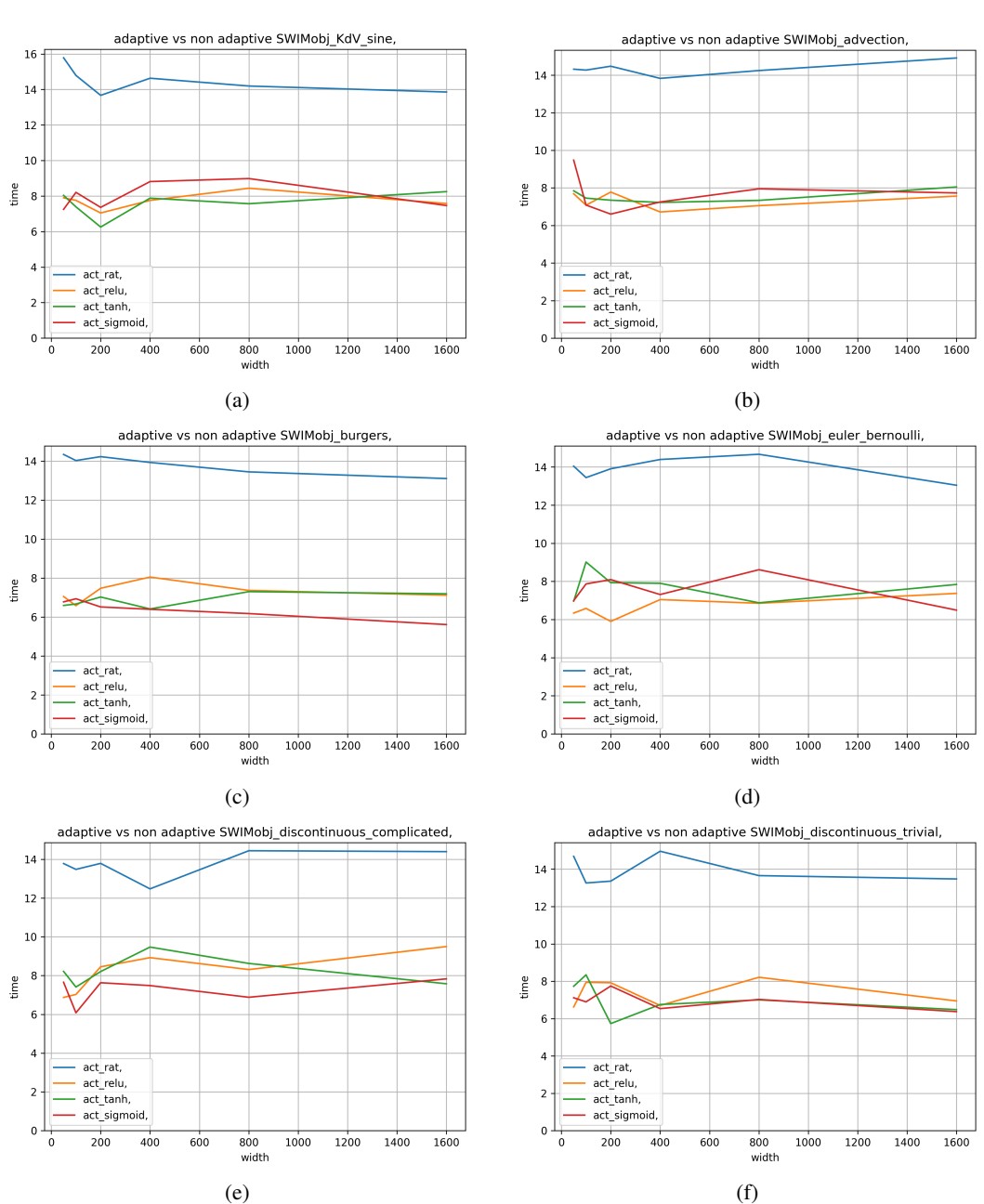

Figure 6: Time a-SWIM vs. SWIM

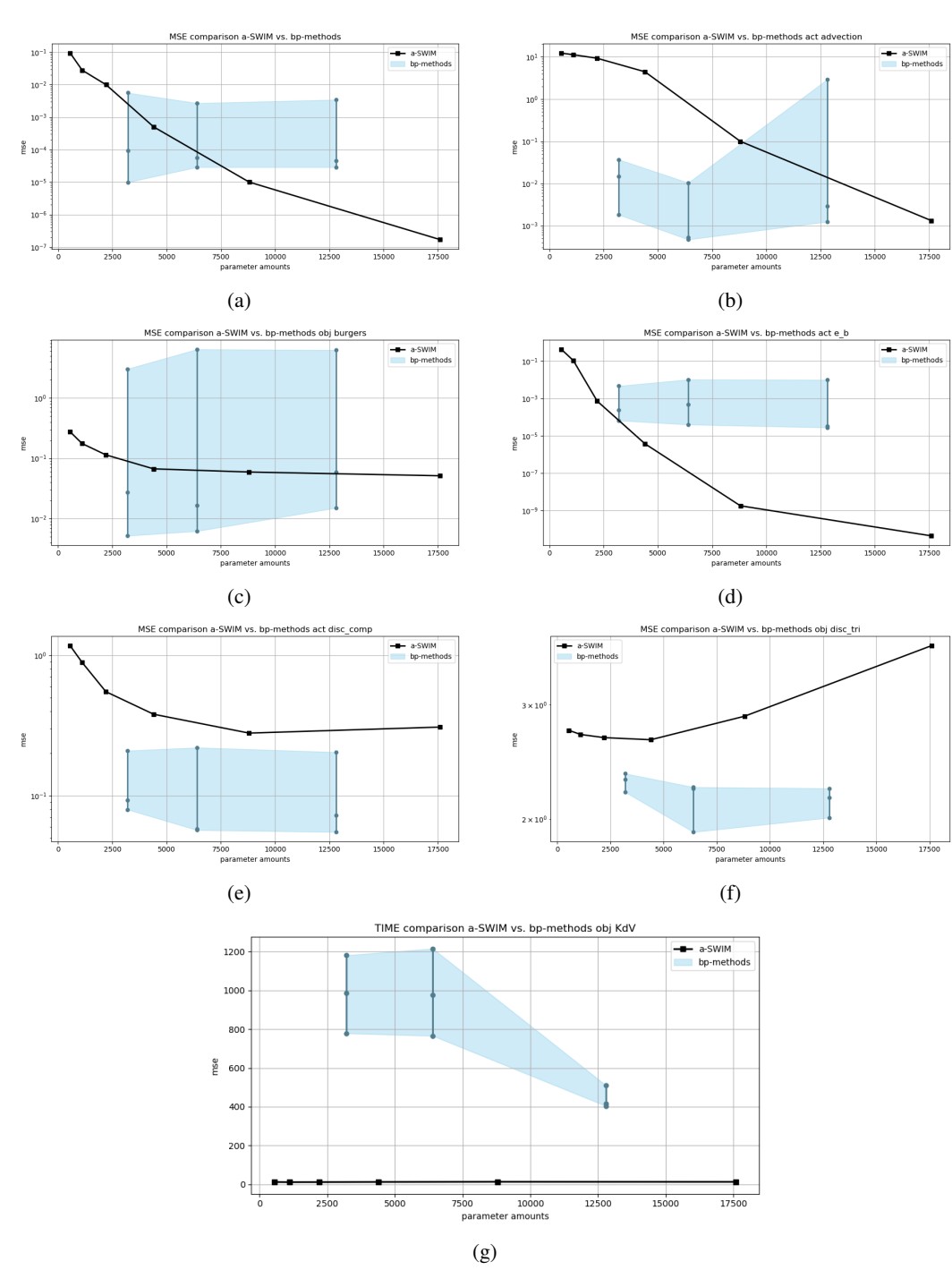

Figure 7: MSE loss a-SWIM vs. BP-NN

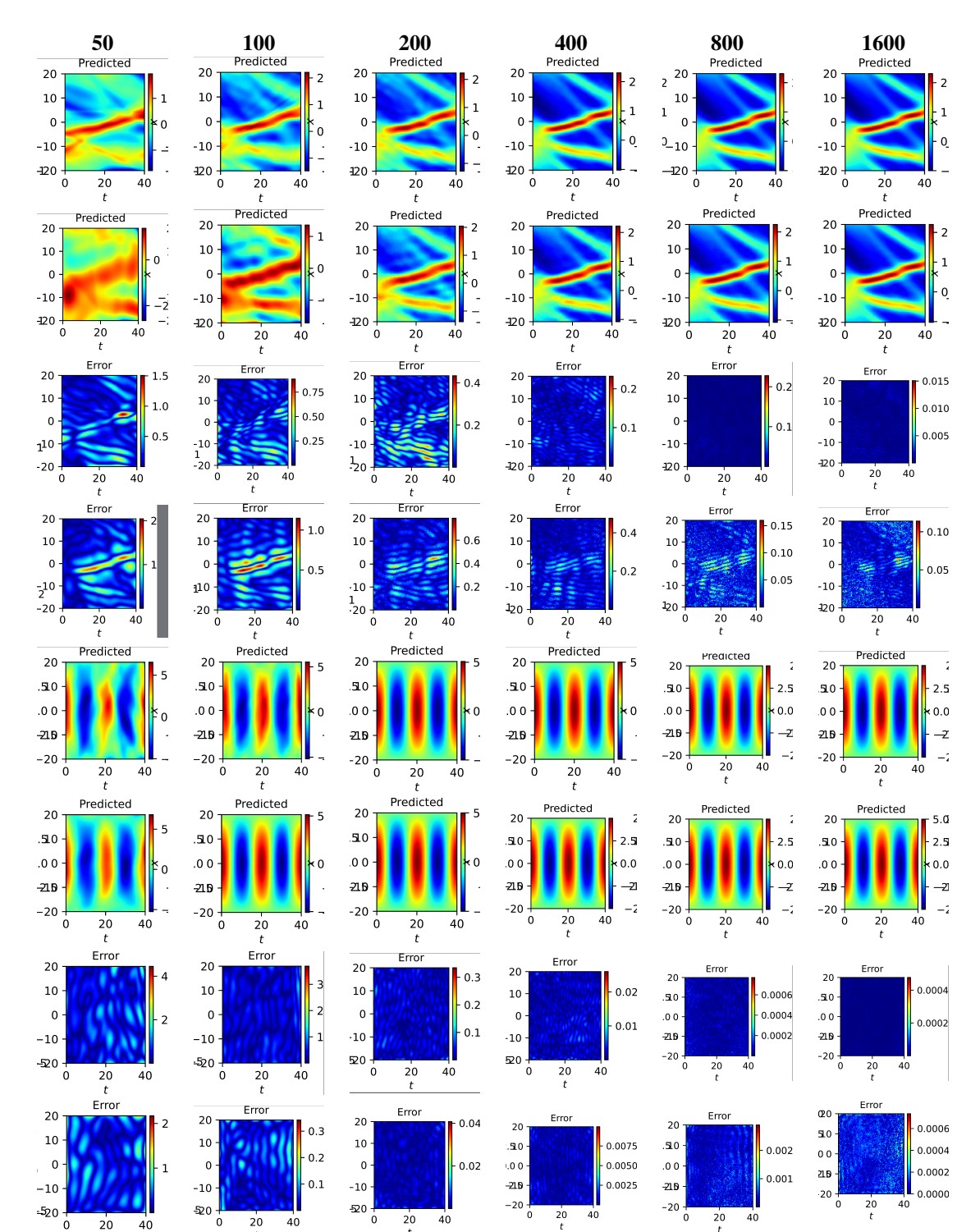

Figure 8: Predicted Dynamics a-SWIM vs. SWIM (Part 1)

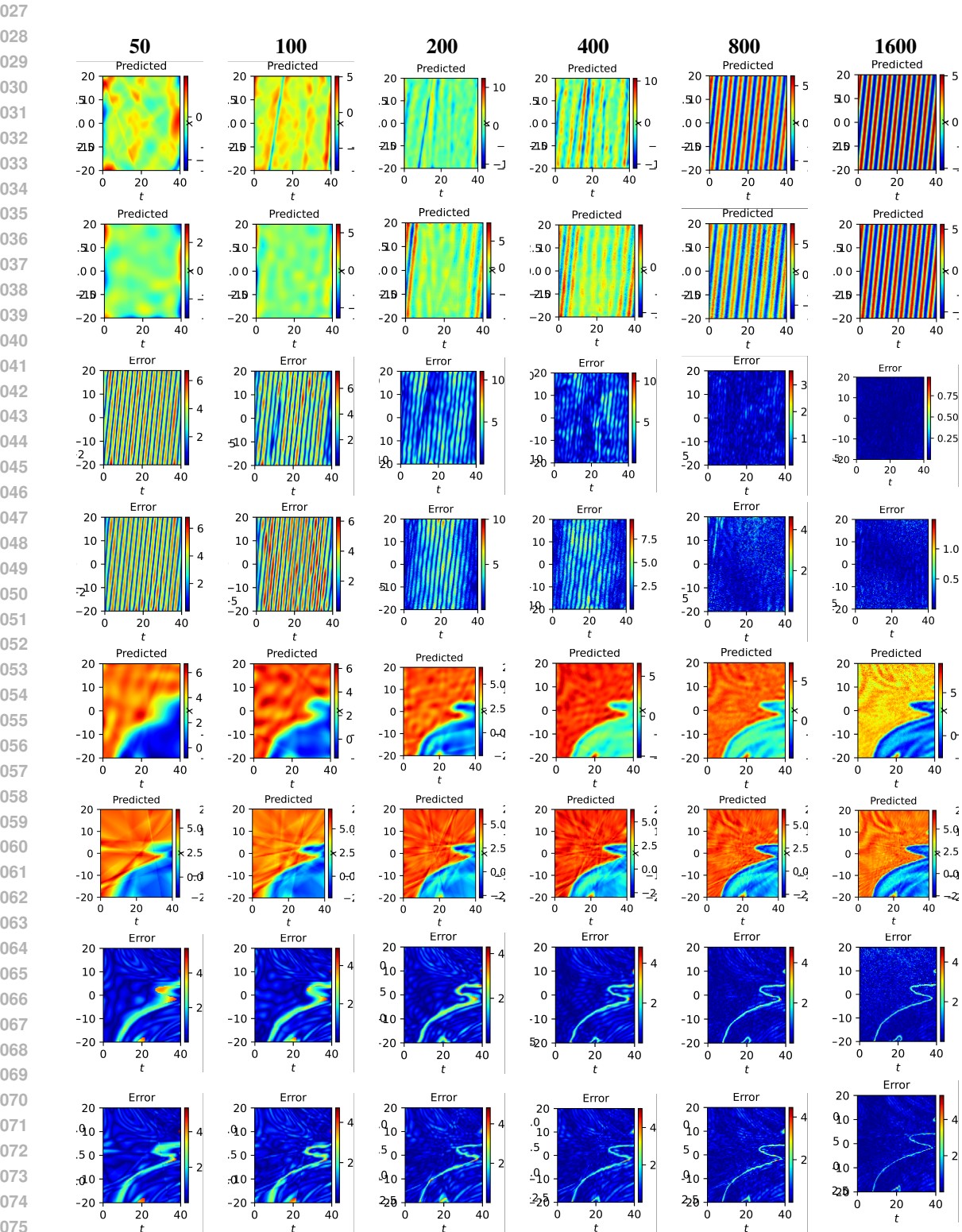

Figure 9: Predicted Dynamics a-SWIM vs. SWIM (Part 2)

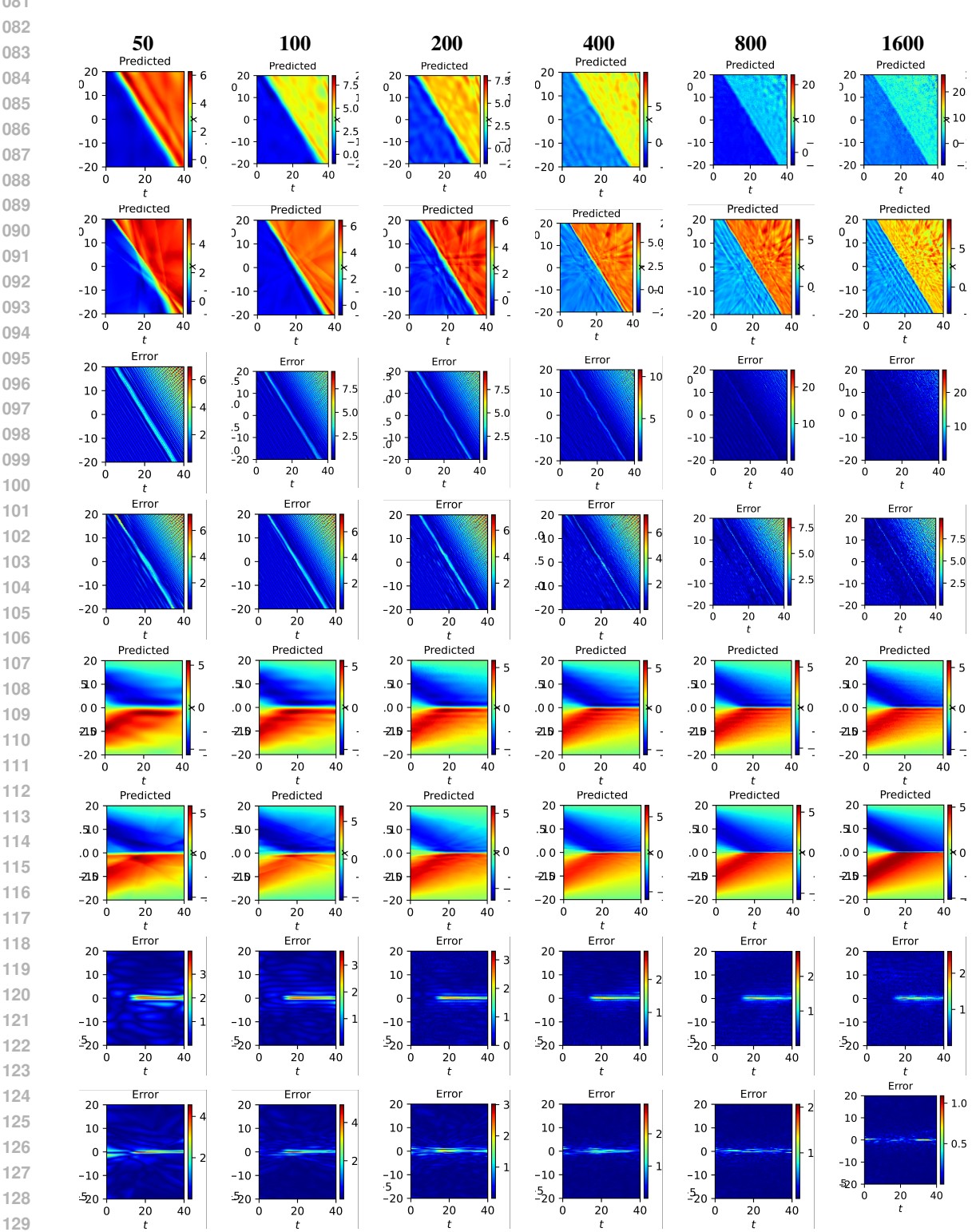

Figure 10: Predicted Dynamics a-SWIM vs. SWIM (Part 3)

Table 1: Experiment statistics Part 1

(a) MSE Losses Obj. KdV_sine

| Width | a-SWIM | ReLU-SWIM | Tanh-SWIM | Sigmoid-SWIM |
|-------|--------|-----------|-----------|--------------|
| 50 | **9.44e-2** | 1.76e-1 | 2.09e-1 | 2.04e-1 |
| 100 | **2.85e-2** | 8.33e-2 | 8.20e-2 | 9.81e-2 |
| 200 | **1.04e-2** | 5.77e-2 | 1.82e-2 | 4.55e-2 |
| 400 | **5.03e-4** | 1.78e-2 | 3.86e-3 | 1.67e-2 |
| 800 | **1.07e-5** | 3.61e-3 | 9.50e-4 | 1.38e-2 |
| 1600 | **1.70e-7** | 1.34e-3 | 3.67e-4 | 3.39e-3 |

(b) MSE Losses Obj. Advection

| Width | a-SWIM | ReLU-SWIM | Tanh-SWIM | Sigmoid-SWIM |
|-------|--------|-----------|-----------|--------------|
| 50 | 12.36 | 12.35 | 12.26 | **12.11** |
| 100 | **11.34** | 12.05 | 11.86 | 11.73 |
| 200 | 9.33 | 9.23 | **8.61** | 10.80 |
| 400 | **4.47** | 5.35 | 6.37 | 8.90 |
| 800 | **1.0e-1** | 2.23 | 4.36e-1 | 7.93 |
| 1600 | **1.32e-3** | 1.17 | 2.29e-2 | 8.98 |

(c) MSE Losses, Obj. E. B.

| Width | a-SWIM | ReLU-SWIM | Tanh-SWIM | Sigmoid-SWIM |
|-------|--------|-----------|-----------|--------------|
| 50 | 4.31e-1 | 7.53e-1 | 1.67e-1 | **5.38e-2** |
| 100 | 1.08e-1 | 2.97e-1 | 3.04e-3 | **7.31e-4** |
| 200 | 7.30e-4 | 4.51e-2 | **6.44e-6** | 8.24e-6 |
| 400 | 3.81e-6 | 1.87e-1 | **4.24e-7** | 5.86e-7 |
| 800 | **1.83e-9** | 8.01e-3 | 1.14e-7 | 2.35e-7 |
| 1600 | **4.48e-11** | 2.17e-3 | 1.49e-8 | 7.97e-8 |

(d) MSE Losses Obj. Burgers

| Width | a-SWIM | ReLU-SWIM | Tanh-SWIM | Sigmoid-SWIM |
|-------|--------|-----------|-----------|--------------|
| 50 | 2.77e-01 | **2.30e-01** | 4.30e-01 | 5.80e-01 |
| 100 | 1.77e-01 | **1.65e-01** | 2.49e-01 | 4.30e-01 |
| 200 | 1.15e-01 | **7.12e-02** | 1.98e-01 | 3.18e-01 |
| 400 | 6.68e-02 | **4.86e-02** | 1.33e-01 | 2.89e-01 |
| 800 | 5.92e-02 | **1.69e-02** | 1.32e-01 | 2.00e-01 |
| 1600 | 5.13e-02 | **3.43e-03** | 5.99e-01 | 2.84e-01 |

(e) MSE Losses Obj. Disc. Tri.

| Width | a-SWIM | ReLU-SWIM | Tanh-SWIM | Sigmoid-SWIM |
|-------|--------|-----------|-----------|--------------|
| 50 | **2.74** | 2.77 | 2.85 | 2.87 |
| 100 | 2.70 | **2.63** | 2.72 | 2.72 |
| 200 | 2.67 | **2.61** | 2.67 | 2.75 |
| 400 | 2.65 | **2.51** | 2.66 | 2.77 |
| 800 | 2.88 | **2.50** | 3.57 | 3.38 |
| 1600 | 3.70 | **2.58** | 4.04 | 9.78 |

(f) MSE Losses Obj. Disc. Comp.

| Width | a-SWIM | ReLU-SWIM | Tanh-SWIM | Sigmoid-SWIM |
|-------|--------|-----------|-----------|--------------|
| 50 | 1.17 | **8.74e-01** | 1.08 | 1.20 |
| 100 | 8.89e-01 | **6.85e-01** | 7.03e-01 | 9.17e-01 |
| 200 | 5.56e-01 | **4.69e-01** | 5.70e-01 | 8.54e-01 |
| 400 | 3.88e-01 | **3.80e-01** | 4.62e-01 | 6.56e-01 |
| 800 | 2.80e-01 | **2.59e-01** | 4.69e-01 | 8.65e-01 |
| 1600 | 3.08e-01 | **1.92e-01** | 4.06e-01 | 1.40 |

Table 2: Experiment statistics Part 2

(a) Training Time of a-SWIM vs. SWIM (in seconds)

| Width | a-SWIM | ReLU-SWIM | Tanh-SWIM | Sigmoid-SWIM |
|-------|--------|-----------|-----------|--------------|
| 50    | 12.30  | 7.90      | 8.04      | 7.25         |
| 100   | 11.66  | 7.75      | 7.39      | 8.21         |
| 200   | 11.96  | 7.05      | 6.25      | 7.36         |
| 400   | 12.58  | 7.75      | 7.88      | 8.81         |
| 800   | 13.39  | 8.44      | 7.57      | 8.98         |
| 1600  | 13.04  | 7.58      | 8.25      | 7.46         |

(b) Training Time of BP-NN (in seconds)

|         | Amount A | Amount B | Amount C |
|---------|----------|----------|----------|
| Shape s | 778.63   | 765.49   | 414.67   |
| Shape m | 984.73   | 976.30   | 403.57   |
| Shape d | 1180.37  | 1214.65  | 510.06   |

Table 3: Notations

| ITEMS | NOTATIONS | ITEMS | NOTATIONS |
|-------|-----------|-------|-----------|
| number of candidates | $N$ | number of neurons | $\mathbf{M}$ |
| number of interpolations | $K$ | input dimension | $d$ |
| repetition factor | $r$ | x-point pairs | $(\mathbf{x}_n^{(s)}, \mathbf{x}_n^{(e)})$ |
| x-point sets | $(\mathbf{x}_n^{(s)}, ..., \mathbf{x}_n^{(k)}, ..., \mathbf{x}_n^{(e)})$ | xu-point sets | $(\mathbf{x}_n^{(s)}, ..., \mathbf{x}_n^{(k)}, ..., \mathbf{x}_n^{(e)})$ $(u_n^{(s)}, ..., u_n^{(k)}, ..., u_n^{(e)})$ |
| candidate weights and biases | $\mathbf{w}_n, b_n$ | candidate adaptive parameters | $\boldsymbol{\theta}_n$ |
| candidate activation functions | $\sigma_{\boldsymbol{\theta}_n}(\cdot)$ | selected weights and biases | $\mathbf{w}_m, b_m$ |
| selected adaptive parameters | $\boldsymbol{\theta}_m$ | selected activation functions | $\sigma_{\boldsymbol{\theta}_m}(\cdot)$ |
| output weights and biases | $\mathbf{w}_{out}, b_{out}$ | scale factors | $s_1, s_2$ |
| dataset size total, train, validation | $C, T, V$ | input total, train, validation | $\mathbf{X}, \mathbf{X}_{trian}, \mathbf{X}_{val}$ |
| input instances | $\mathbf{x}, \mathbf{x}_{trian}, \mathbf{x}_{val}$ | output total, train, validation | $\mathbf{u}, \mathbf{u}_{trian}, \mathbf{u}_{val}$ |
| output instances | $u, u_{trian}, u_{val}$ | objective function | $u = u(\mathbf{x})$ |
| numerator, denominator degree | $r_p, r_q$ | numerator. denominator polynomials | $P(\cdot), Q(\cdot)$ |
| numerator coefficients | $\alpha^{(i)}$ | denominator coefficients | $\beta^{(i)}$ |

