# OpenReview forum: "Neural Networks with Adaptive Activation Functions and their Application to the Solution of PDEs"
_ICLR.cc/2026/Conference — ICLR 2026 Conference Withdrawn Submission_

### Official Review · Reviewer_w23h · 2025-10-17

**Soundness:** 1
**Presentation:** 1
**Contribution:** 1
**Rating:** 2
**Confidence:** 4

**Summary:**

This work aims to approximate solutions to partial differential equations (PDEs) by fully connected neural networks. The authors propose a method that extends a recent sampling-based approach to replace backpropagation with a forward sampling scheme to accelerate training. The main difference from previous works is the use of adaptive activation functions. The algorithm is applied on a range of solutions to two-dimensional PDEs and achieved competitive mean squared error compared with SWIM with a range of activation functions.

**Strengths:**

- The problem of solving high-dimensional PDEs by neural networks is timely and there is a rich literature on this topic (e.g. physics-informed neural networks, PINNs).
- The idea of using adaptive activation functions is interesting and has the potential to improve the expressiveness of the neural networks an has been explored for PDEs by Jagtap et al. (2020).

**Weaknesses:**

- The paper is not well-written and several sections are difficult to follow with not rigorous definitions. See for instance, the paragraph on ground truth dynamics, Eq.~(12) where the parameters are not introduced, or Eq. (17) where the probability distribution is not clearly defined.
- There is a significant difference between the claims and motivations of the paper and the actual experiments. First, the definition of the numerical task is wrong. Usually, one either solves a PDE by minimizing the residual with a neural network (as in PINNs); otherwise why would we construct a neural network approximation if we already know the solution? Or, one uses pairs of source terms and solutions to train a neural operator to approximation the solution map associated with a PDE to evaluate it at new source terms. However, in the experiments, the authors simply fit a neural network to a known solution without any source term. This is not representative of the actual task of solving PDEs. Second, the experiments are limited to low-dimensional PDEs (1D and 2D), while the main motivation of using neural networks for PDEs is to tackle high-dimensional problems where traditional numerical methods fail.
- The novelty of the proposed method is limited, as it mainly combines existing techniques (sampling-based training and adaptive activation functions) without significant new contributions.

**Questions:**

- How does the proposed method compare to standard PINNs or other neural operator methods in terms of accuracy and computational efficiency?
- How does the method scale to high-dimensional PDEs, which is one of the main motivations for using neural networks in this context? In this case, I am worried that the sampling-based approach may become inefficient due to the curse of dimensionality (there would be no intermediate points to sample in the dataset from Eq. (6)).
- Is there any approximation theory (beyond universal approximation theorem) that supports the use of adaptive SWIM over standard SWIM?

---

### Official Review · Reviewer_NWRf · 2025-10-17

**Soundness:** 2
**Presentation:** 2
**Contribution:** 2
**Rating:** 2
**Confidence:** 3

**Summary:**

This paper proposes a framework that combines sampling-based training (i.e., the SWIM method) with adaptive activation functions (specifically, a modified rational function derived from a variant of the Pade Approximation Unit). The authors’ main goal is to improve the efficiency when approximating functions, with a particular emphasis on solving partial differential equations (PDEs). The method avoids backpropagation by solving a series of small sub-optimization problems (one for each candidate neuron) and then selects neurons based on various probability strategies. A thorough experimental evaluation is provided, comparing a-SWIM with both non-adaptive SWIM variants and standard backpropagation-trained neural networks on several objective functions

**Strengths:**

- Novel Integration:
The paper successfully blends two methodologies—sampling-based training (SWIM) and adaptive activation functions—to create an approach that can potentially combine the efficiency of sampling with the expressive power of trainable activations.
- Comprehensive Experiments: The experimental evaluation seems extensive. The authors compare a-SWIM not only with various fixed-activation SWIM versions (using ReLU, Tanh, Sigmoid) but also with BP-trained networks. Multiple PDEs and objective functions—covering smooth, discontinuous, and high-frequency cases—are used. The analysis covers both prediction accuracy (MSE) and training time.

**Weaknesses:**

- **The presentation throughout the paper could be improved for clarity.** (i) For example, in Section 3, the formulation of the PDE solution mapping $u(x)$ as a scalar-valued function is not well justified—it would be beneficial to explain how the method could be extended to handle multi-valued function cases or why we just consider scalar-valued functions. (ii) In Section 3 (Background), line 89, the sentence in question appears to delve into experimental specifics. Typically, the background section is reserved for introducing the general problem and relevant context rather than experimental details. It might be more appropriate to relocate this sentence to the experiments section. (iii) Besides, Section 3 assumes a supervised training setting with ground truth training data available; however, for many complex PDEs, such as those without analytical solutions, ground truth data is often not available. This disconnect between the motivation and the chosen methodology requires further clarification. (iv) Section 4 (Methods) is hard to follow. I think it would be better to first briefly recall the methodology of existing methods (SWIM and adaptive neural networks). Next, give the pipeline of your approach (like Fig. 3 the right one). Finally, provide the details of your approach step by step. Otherwise, for example, readers may not understand why you present 4.1 XU-Point Sets first.
-  **Lack of Theoretical Justification**. While the paper provides a detailed derivation of the method, the theoretical analysis supporting the benefits of adaptive activation functions in a sampling-based regime remains somewhat informal. Concretely, can a generalization bound of your model derived from some results from existing work or what is the approximation capability of the proposed NN model. I understand this is a paper in application theory. However, a more rigorous theoretical discussion or error bound analysis, even if approximate, would strengthen the contribution.
- **No Scalability Demonstration to Higher-Dimensional Problems**:
The experiments in the paper are conducted solely for objective functions with an input dimension of
$d=2$. This limited scope raises questions about the generality and scalability of the proposed framework. Without experimental evidence or discussion on higher-dimensional scenarios, it remains uncertain whether the method would perform adequately for problems with input dimensions greater than 2—a common occurrence in real-world PDE applications.

**Questions:**

See Weakness Part, and the following are additional questions.

- In 4.1 XU-Point Sets, you chose x-point pairs in eq. (3). What's the relationship of $x^{(s)}$ and $x^{(e)}$ in each pair or should they satisfy some conditions?
- It would be better to put some key results of the experiments to the main body of the content, which can enhance the expressivity of this paper.
- the experiments seem to be conducted only for the case where $d=2$. Could the authors elaborate on how the proposed framework is expected to perform or scale when dealing with higher-dimensional problems?

Minor:
- In eq. (5), there is a typo for $(x_n^{(s)},x^{(1)}_nx^{(2)}_n...)$?

**Details Of Ethics Concerns:**

No Concerns.

---

### Official Review · Reviewer_LyDC · 2025-10-27

**Soundness:** 2
**Presentation:** 1
**Contribution:** 1
**Rating:** 0
**Confidence:** 4

**Summary:**

This paper develops a method for solving PDEs using rational activation functions which are learned and randomly sampled neural networks using a variant of SWIM. My main problem with the paper is that it appears throw together many different unrelated ideas to construct a method. There are also no theoretical contributions in the paper. The resulting method is very complicated and in my opinion unmotivated. All of the experiments in the paper have been moved to the appendix. I would recommend the authors pick one or two experiments which most convincingly show the improvement of their method over other approaches and add these to the main text. For these reasons, I unfortunately recommend the paper in its present form be rejected.

**Strengths:**

The experiments shown in the appendix do look rather good, but it is difficult to get an overview over what is happening in the 7 different experiments and what they are supposed to be showing.

**Weaknesses:**

The paper lacks theoretical analysis and appears to consists of a very complicated method obtained by combining numerous existing ideas. For this reason, I question the novelty of the paper. I'm not convinced that there are any significant new ideas, although certainly the authors have done a good job engineering a solid method.

The presentation of the paper is very difficult to understand. Too many different things are combined together in an unmotivated way.

**Questions:**

Why is the domain $\mathcal{D}$ typically $[-20,20]\times [0,40]$ when solving PDEs? I suppose these are what you considered in your experiments.

It is not clear to me how the activation function in equation (16) was obtained. Is this the best activation function for all of the problems you considered?

---

### Note · Authors · 2025-12-22

I have read and agree with the venue's withdrawal policy on behalf of myself and my co-authors.